# New Findings: Hindlimb Unloading Causes Nucleocytoplasmic Ca^2+^ Overload and DNA Damage in Skeletal Muscle

**DOI:** 10.3390/cells12071077

**Published:** 2023-04-03

**Authors:** Huajian Yang, Huiping Wang, Fangyang Pan, Yuxi Guo, Liqi Cao, Wenjing Yan, Yunfang Gao

**Affiliations:** 1Shaanxi Key Laboratory for Animal Conservation, College of Life Sciences, Northwest University, Xi’an 710069, China; 202021095@stumail.nwu.edu.cn (H.Y.); wanghp@nwu.edu.cn (H.W.); 202121589@stumail.nwu.edu.cn (F.P.); 202233103@stumail.nwu.edu.cn (Y.G.); 202233066@stumail.nwu.edu.cn (L.C.); 202221258@stumail.nwu.edu.cn (W.Y.); 2Key Laboratory of Resource Biology and Biotechnology in Western China, College of Life Sciences, Northwest University, Ministry of Education, Xi’an 710069, China

**Keywords:** skeletal muscle, hindlimb unloading, nuclear Ca^2+^ regulation, nuclear apoptosis

## Abstract

Disuse atrophy of skeletal muscle is associated with a severe imbalance in cellular Ca^2+^ homeostasis and marked increase in nuclear apoptosis. Nuclear Ca^2+^ is involved in the regulation of cellular Ca^2+^ homeostasis. However, it remains unclear whether nuclear Ca^2+^ levels change under skeletal muscle disuse conditions, and whether changes in nuclear Ca^2+^ levels are associated with nuclear apoptosis. In this study, changes in Ca^2+^ levels, Ca^2+^ transporters, and regulatory factors in the nucleus of hindlimb unloaded rat soleus muscle were examined to investigate the effects of disuse on nuclear Ca^2+^ homeostasis and apoptosis. Results showed that, after hindlimb unloading, the nuclear envelope Ca^2+^ levels ([Ca^2+^]_NE_) and nucleocytoplasmic Ca^2+^ levels ([Ca^2+^]_NC_) increased by 78% (*p* < 0.01) and 106% (*p* < 0.01), respectively. The levels of Ca^2+^-ATPase type 2 (Ca^2+^-ATPase2), Ryanodine receptor 1 (RyR1), Inositol 1,4,5-tetrakisphosphate receptor 1 (IP_3_R1), Cyclic ADP ribose hydrolase (CD38) and Inositol 1,4,5-tetrakisphosphate (IP_3_) increased by 470% (*p* < 0.001), 94% (*p* < 0.05), 170% (*p* < 0.001), 640% (*p* < 0.001) and 12% (*p* < 0.05), respectively, and the levels of Na^+^/Ca^2+^ exchanger 3 (NCX3), Ca^2+^/calmodulin dependent protein kinase II (CaMK II) and Protein kinase A (PKA) decreased by 54% (*p* < 0.001), 33% (*p* < 0.05) and 5% (*p* > 0.05), respectively. In addition, DNase X is mainly localized in the myonucleus and its activity is elevated after hindlimb unloading. Overall, our results suggest that enhanced Ca^2+^ uptake from cytoplasm is involved in the increase in [Ca^2+^]_NE_ after hindlimb unloading. Moreover, the increase in [Ca^2+^]_NC_ is attributed to increased Ca^2+^ release into nucleocytoplasm and weakened Ca^2+^ uptake from nucleocytoplasm. DNase X is activated due to elevated [Ca^2+^]_NC_, leading to DNA fragmentation in myonucleus, ultimately initiating myonuclear apoptosis. Nucleocytoplasmic Ca^2+^ overload may contribute to the increased incidence of myonuclear apoptosis in disused skeletal muscle.

## 1. Introduction

Disuse conditions, such as spaceflight, prolonged bed rest, and limb breakage, can induce skeletal muscle atrophy, which is characterized by muscle mass loss, cross-sectional area (CSA) reduction, skeletal muscle proteins degradation, and muscle fiber type transformation, resulting in motor function impairing and metabolic disease risk significant increasing [1,2].

A severe imbalance in cytosolic Ca^2+^ homeostasis occurs during disuse atrophy of skeletal muscle manifested as cytoplasmic Ca^2+^ overload [3,4,5]. As an important organelle, the nucleus exerts its own unique functions and serves as an important Ca^2+^ reservoir in the regulation of cellular Ca^2+^ homeostasis [6]. The nucleus consists of an outer (ONM) and inner nuclear membrane (INM), which fuse at the nuclear pore complex [7]. The lumen between the ONM and INM, known as the nuclear envelope, directly communicates with the lumen of the endoplasmic reticulum for Ca^2+^ storage. Nuclear Ca^2+^ levels are primarily regulated by Ca^2+^ transporters in the nuclear membrane. Ca^2+^ communication between the cytoplasm and nuclear envelope and between the nuclear envelope and nucleocytoplasm occurs via Ca^2+^ transporters. Ca^2+^ levels in the nuclear envelope and the capacity of Ca^2+^ transporters are essential for maintaining nucleocytoplasmic Ca^2+^ homeostasis [8,9]. Currently, however, changes in Ca^2+^ levels in the nuclear envelope and nucleocytoplasm and changes in Ca^2+^ transporters under conditions of hindlimb unloading are unknown.

Nuclear structure and function are regulated by Ca^2+^ levels in the nucleus and various biological activities, such as gene transcription [10,11], cell cycle regulation [12], DNA synthesis and repair [13], and nuclear membrane rupture [14], are closely related to nuclear Ca^2+^. Nucleocytoplasmic Ca^2+^ overload can cause nuclear apoptosis [15], and activation of Ca^2+^-dependent deoxyribonuclease (DNase) in skeletal muscle can lead to DNA breakage and nuclear apoptosis [16]. Disuse muscle atrophy is accompanied by a marked increase in nuclear apoptosis in skeletal muscle [17,18]. However, whether nuclear apoptosis in disuse skeletal muscle atrophy is associated with DNase activation by nucleocytoplasmic Ca^2+^ overload is unclear.

In the present study, we explored Ca^2+^ levels in the nuclear envelope and nucleocytoplasm ([Ca^2+^]_NE_ and [Ca^2+^]_NC_) in hindlimb-unloaded rat skeletal muscle and examined the capacity and regulation of Ca^2+^ transporters on nuclear membranes to identify possible reasons for changes in nuclear Ca^2+^ levels in atrophied muscle. We also explored the potential relationship between nucleocytoplasmic Ca^2+^ and nuclear apoptosis in atrophied muscle. Therefore, we investigated dysregulation of nuclear Ca^2+^ homeostasis in disused skeletal muscle to clarify the potential mechanisms and effects of intracellular Ca^2+^ disturbance in disused skeletal muscle, thereby providing novel targets for the prevention and treatment of disuse atrophy in skeletal muscle.

## 2. Materials and Methods

### 2.1. Hindlimb Unloading Animal Modeling and Sample Preparation

All procedures were approved by the Laboratory animal care committee of the P. R. China Ministry of Health. Sprague-Dawley adult healthy female rats, weighing 220−240 g, were provided by Dashuo Company (Chengdu, China). The individuals were randomly divided into two groups: cage control (CON) group and hindlimb unloading (HLU) group. The CON individual could move freely in a normal cage. The HLU individual was tail suspended for 14 days. The suspending method was referred to Morey-Holton [19] and modified. Briefly, the individual tail was washed and wiped with benzoin and rosin. After air drying, the tail was evenly wrapped with gauze. Then, the tail was linked with medical adhesive tape to a rotatable swivel joint which allowed 360° rotation in the top of the suspension cage. The individual was kept at an inclination of approximately 30° in a head-down orientation and allowed to move freely in the cage on its front feet. The size of the suspension cage is 30 cm × 35 cm × 45 cm. All individuals were separately housed in the single cage with a 12:12 h light-dark cycle and 20.0 ± 2.0 °C environment and fed standard laboratory rat chow and water ad libitum.

At the sampling time, the individuals were anesthetized via intraperitoneal administration of urethane (1000 mg/kg). The soleus (SOL) muscle was carefully separated and quickly surgically removed. The isolated muscle sample was immediately used for [Ca^2+^] measurement, WB, ELISA and DNA fragmentation assay, or fixed in 4% paraformaldehyde for TUNEL assay and immunofluorescence staining. After surgery, the individuals were euthanized by injecting an overdose of urethane.

### 2.2. Nuclei Isolation and [Ca^2+^]_NE_ and [Ca^2+^]_NC_ Measurement 

Nuclei isolation was performed according to the method of Li, [20] with some modifications. Fresh muscle sample was minced into 0.5 mL Buffer A (2 mM MgCl_2_, 125 mM KCl, 10 mM HEPES, protease inhibitors), then placed quietly on ice for 10 min and homogenized. The homogenate was filtered through a 200-mesh cell sieve. After centrifugation (1000× *g*, 10 min, 4 °C), the precipitate was resuspended in 0.5 mL Buffer B (0.25 M sucrose in buffer A). The solution was gently poured onto the top of 0.5 mL Buffer C (1.1 M sucrose in buffer A) and centrifuged (1000× *g*, 10 min, 4 °C). The nuclei were enriched at the bottom. The nuclei were resuspended in Hank’s Balanced Salt Solution (HBSS) without Ca^2+^ (G4203, Servicebio, Wuhan, China). The resuspended nuclei were stained with 1% DAPI (D9542, Sigma-Aldrich, St. Louis, MI, USA) to detect the volume and density of the isolated nuclei and examined for nuclear integrity under bright field by laser confocal microscope (TCS SP8, Leica, Wetzlar, Germany).

The resuspended nuclei were divided into two parts. One part was used for [Ca^2+^]_NE_ measurement, incubating with 20 μM Cal-520 AM (No. 21130, AAT Bioquest, Los Angeles, CA, USA) containing 0.04% Pluronic F-127 (ST501, Beyotime, Shanghai, China) for 20 min, 37 °C. Other part was used for [Ca^2+^]_NC_ measurement, incubating with 30 μg/mL Cal-520 Dextran (No. 20600, AAT Bioquest, Los Angeles, CA, USA) for 20 min, 25 °C. The nucleus and [Ca^2+^] fluorescence were imaged by laser confocal microscope (TCS SP8, Leica, Wetzlar, Germany). The [Ca^2+^] fluorescence intensity was measured by Fiji-ImageJ software (National Institutes of Health, Bethesda, MD, USA). Mean fluorescence (F) and background value (F_0_) were measured at a given region of interest (ROI). The content of [Ca^2+^] was represented by the relative fluorescence intensity (F-F_0_).

### 2.3. Western Blot Assay (WB)

#### 2.3.1. Nuclear Protein Extraction

The muscle nuclei were isolated using Nuclei Isolate Kit (SN0020, Solarbio, Beijing, China). The steps were as follows: the muscle sample was minced into 1.0 mL pre-cooled Lysis Buffer containing 50 μL Reagent A. It was then homogenized on ice and filtered through a 200-mesh cell sieve. After centrifugation (700× *g*, 5 min, 4 °C), the precipitate was resuspended in 0.5 mL pre-cooled Lysis Buffer. Then, the solution was gently poured onto the top of 0.5 mL Medium Buffer and centrifuged (700× *g*, 5 min, 4 °C). The precipitate was resuspended in 0.5 mL Lysis Buffer again and centrifuged (1000× *g*, 10 min, 4 °C). The nuclei were enriched in the bottom. Nuclear protein was extracted according to Wilkie’s method [21]. The nuclei were dissolved in 50 μL PBS contain 0.1% Triton X-100 and 6 M urea. After centrifugation (6000× *g*, 2 min, 4 °C), the supernatant was collected. The protein concentration in the supernatant was measured by the Bicinchoninic acid (BCA) kit (AR0197B, Boster, Wuhan, China).

#### 2.3.2. Western Blot

The supernatant were mixed with 1 × SDS loading buffer (1:4 v:v), boiled in a metal bath and stored at −20 °C for further analysis. Nuclear proteins were separated by SDS-PAGE (8% Laemmli gel, acrylamide/bisacrylamide ratio 29:1, 98% 2,2,2-trichloroethanol), and each lane contains 10 μg of protein. After electrophoresis, proteins were electrically transferred onto a PVDF membrane (0.22 µm, Merck Millipore, Burlington, MA, USA) using a Bio-Rad wet transfer device (Hercules, CA, USA). The membrane was blocked with 5% skim milk in TBST (10 mM Tris-HCl, 150 mM NaCl, 0.1% Tween-20, pH 7.6) for 1 h, and incubated with primary antibody in TBST at 4 °C for overnight. The primary antibodies and their dilutions were listed in Table 1. After washed with TBST, the membrane was incubated with goat anti-Rabbit IgG (H + L) secondary antibody (1:2000, EK020, Zhuangzhi Bio, Xi’an, China) for 90 min at room temperature. After washing with TBST, the hybridization band would be luminous with chemiluminescence reagent West Pico Plus (No. 34580, Thermo Fisher Scientific, Waltham, MA, USA), then visualized with a scanner (G: Box, GBOX Cambridge, UK). Quantification analysis of the band was performed using NIH Image J software (Media Cybernetics, Inc., Rockville, MD, USA) and normalized expression as a percentage of the total protein staining of each lane.

### 2.4. IP_3_ and IP_4_ Content Measurement

The content of Inositol 1,4,5-tetrakisphosphate (IP_3_) and Inositol 1,3,4,5-tetrakisphosphate (IP_4_) in SOL was measured by Enzyme linked Immunosorbent assay (ELISA). The SOL muscle was homogenized with PBS (1:9, m:v) on ice. After centrifugation (2000× *g*, 15 min, 4 °C), the supernatant was collected for further measurement. The IP_3_ and IP_4_ concentration in the supernatant was measured according to the instructions of the kits (F40412-B, F40516-B, Fankew, Shanghai, China). The protein concentration in the supernatant was measured by the Bicinchoninic acid (BCA) kit (AR0197B, Boster, Wuhan, China). The IP_3_ and IP_4_ content was normalized expression as the protein content.

### 2.5. TUNEL Staining for Nucleus and Immunofluorescence Staining for DNase X

The apoptosis of nucleus and the involvement of DNase X were determined by TUNEL staining and immunofluorescence staining. Muscle samples which had been fixed in 4% paraformaldehyde for more than 24 h were used for this protocol. The sample was paraffin-sectioned into a slice using conventional methods. The slice was dewaxed in gradient ethanol and repaired by 10% proteinase K (G1205, Servicebio, Wuhan, China). The slice was then permeabilized with 0.1% Triton X-100 at room temperature for 20 min. TUNEL staining was strictly carried out according to the manufacturer’s instruction in the kit (G1502, Servicebio, Wuhan, China). Briefly, the mixed solution (TdT enzyme, dUTP, buffer, 1:5:50) was dropped on the slice, which was then incubated at 37 ℃ for 2 h. After washing with PBS three times, the slice was covered with 3% BSA (G5001, Servicebio, Wuhan, China) to block non-specific binding for 30 min. The slice was washed with PBS and then incubated with primary antibody of DNase X (1:100; 13653-1-AP, Proteintech, Wuhan, China) and pericentriolar material 1 (PCM1) (1:600; GB114537, Servicebio, Wuhan, China) overnight at 4 °C. After washing with PBS, the slice was incubated with the fluorochrome-conjugated secondary antibodies (1:100; GB22303, Servicebio, Wuhan, China) at room temperature for 50 min. After washing with PBS, the slice was stained with DAPI (D9542, Sigma−Aldrich, St. Louis, MI, USA) in the dark at room temperature for 10 min to present the nucleus. Finally, the slice was observed, and the image was captured using a laser confocal microscope (TCS SP8, Leica, Wetzlar, Germany). Fiji-ImageJ software (National Institutes of Health, Bethesda, MD, USA) was used to count the total number of nuclei, as well as the number of TUNEL-positive nuclei, TUNEL-positive nuclei co-localized with PCM1, and TUNEL-positive nuclei co-localized with DNase X in the slice. The quantitative analysis results were expressed as a percentage of TUNEL-positive nuclei to total nuclei, and TUNEL-positive nuclei co-localized with PCM1 or DNase X to TUNEL-positive nuclei.

### 2.6. DNA Fragmentation Assay

The activity of DNase X was expressed by the DNA fragmentation assay. The DNA fragmentation was determined using plasmid analysis as described in Shiokawa’s report [22]. About 500 mg fresh sample tissue was minced into 300 μL extraction buffer (20 mM Tris-HCl pH 7.8, 1 mM β-mertoethyl alcohol, 300 mM NaCl, 3 mM MgCl_2_, 0.5% Triton X-100, protease inhibitor), then homogenized and placed quietly on ice for 30 min. After centrifugation (10,000× *g*, 10 min, 4 °C), the supernatant was collected. The protein concentration in the supernatant was measured by the Bicinchoninic acid (BCA) kit (AR0197B, Boster, Wuhan, China). An amount of supernatant containing 4 μg protein was mixed into 10 μL reaction solution (50 mM HEPES-NaOH pH 7.2, 1 mM 2-mercaptoethanol) containing 500 ng supercoiled pBluescript II KS+, and incubated for digesting plasmid DNA at 37 °C for 10 min. After that, 10 μL digestion solution was loaded for electrophoresis by 1% agarose gel containing 1 × GeneGreen Nucleic Acid Dye (RT210, Tiangen Biotech, Beijing, China). The gel was visualized with a scanner (G: Box, GBOX Cambridge, UK). Quantification analysis of the band was performed using NIH Image J software (Media Cybernetics Inc., Rockville, MD, USA). The content proportion of DNA fragments below 1000 bp was calculated.

### 2.7. Data Statistics

All statistical tests were performed using SPSS 20 software (IBM, Armonk, NY, USA). Shapiro–Wilk (*p* > 0.05) was used to test the normality distribution of the data. The independent sample Student’s *t*-test was used for comparison between the two groups. *p* < 0.05 was set as the significance level. Data are presented as Mean ± Standard error (SEM). All figures were constructed using GraphPad Prism software (San Diego, CA, USA).

## 3. Results

### 3.1. The Changes of Body Mass and Muscle Morphology

As the animal individuals entered into the experiment (at Entrance), the body mass was 227.2 ± 3.3 and 227.6 ± 4.3 g in CON and HLU group, respectively. When the muscle was sampled after 14 days (at Sampling), the body mass had increased significantly in CON (*p* < 0.01), but slightly in HLU (*p* > 0.05) (Figure 1A). The muscle wet mass and the ratio of muscle mass/body mass of SOL in HLU reduced 60% (*p* < 0.001) and 56% (*p* < 0.001) compared to CON (Figure 1B, C), respectively. The transection image of muscle fiber is presented in Figure 1D. The cross-sectional area (CSA) of muscle fiber in these images presented a 66% decrease in HLU (*p* < 0.001) compared to CON (Figure 1E).

### 3.2. Changes of [Ca^2+^]_NE_ and [Ca^2+^]_NC_

Skeletal muscle tissue contains multiple types of nuclei except for myonucleus. Previous studies have reported that myonucleus comprises 50–60% of total nuclei in rat SOL muscle, while myosatellite, vascular endothelial, smooth muscle and nerve cell nuclei account for approximately 2–5%, 10–20% and 20%, respectively [23]. Morphologically, myonuclei are described as ellipsoidal in shape with a length-to-width ratio of 2.4–4.2 [24]. In our study, we isolated nuclei from SOL muscle and observed them under a microscope, as shown in Figure 2A. The isolated nuclei were stained blue with DAPI and exhibited ellipsoidal, round, slug, and other irregular shapes. Based on the above morphological descriptions of the myonucleus, we primarily targeted the ellipsoidal-shaped nuclei with smooth outer edges and no folds, which accounted for approximately 40% of total nuclei. We then measured the length-to-width ratio of these nuclei and found that nearly 70% had a ratio of 2.5–3.7. Thus, we selected ellipsoidal nuclei with smooth outer edges, no folds, and a length-to-width ratio between 2.5 and 3.7 as the observation targets for myonuclei (as shown in Figure 2B). After loading Cal-520 AM into the selected myonucleus, green fluorescence formed an elliptical ring, indicating that free Ca^2+^ was distributed across the nuclear envelope (Figure 2C). In contrast, after loading Cal-520 Dextran into the same such myonucleus, green fluorescence formed a solid ellipsoid, indicating that free Ca^2+^ was distributed in the nucleocytoplasm (Figure 2D). The negative control for Ca^2+^ fluorescence was presented in Appendix A.

Ca^2+^ fluorescence intensity normally can be used to indicate the free Ca^2+^ content. The Ca^2+^ fluorescence in nuclear envelope and nucleocytoplasm in the CON and HLU groups were shown, respectively (Figure 3A,C). The [Ca^2+^]_NE_ and [Ca^2+^]_NC_, free Ca^2+^ content in nuclear envelope and nucleocytoplasm, in the HLU group significantly increased by 78% (*p* < 0.01) and 106% (*p* < 0.01) compared to the CON group (Figure 3B,D).

In addition, we compared the length, width, and length-to-width ratio of myonucleus in the two groups. In the CON group, the length, width, and ratio were 15.09 ± 0.57 μm, 5.09 ± 0.21 μm, and 2.98 ± 0.11, respectively. In the HLU group, they were 14.92 ± 0.45 μm, 5.08 ± 0.19 μm, and 2.95 ± 0.10, respectively. Thus, there were no significant changes in morphology of the myonucleus after hindlimb unloading.

### 3.3. Relative Expression level of Ca^2+^ Transporters Located on Nuclear Membrane and Its Regulatory Proteins

According to the references [25,26], Calsequestrin1 (CSQ1) and Calnexin (CAX) as endoplasmic reticulum marker protein, and the lamina protein B1 (LaminB1) as nucleus marker protein, were used to determine the purity of nuclear protein extraction. The results showed that CSQ1 and CAX were rich in cellular protein fraction but absent in nuclear protein fraction. Furthermore, LaminB1 was evident in nuclear protein fraction, whereas its presence in the cell protein fraction was minimal (Figure 4A), indicating the isolated nuclei were of high quality. The relative expression levels of Ca^2+^ transporters and regulatory proteins in SOL muscle nuclei in the CON and HLU groups were shown (Figure 4B). In detail, the expression levels of Ryanodine receptor 1 (RyR1), Inositol 1,4,5-tetrakisphosphate receptor 1 (IP_3_R1), Ca^2+^-ATPase type 2 (Ca^2+^-ATPase2), and Cyclic ADP ribose hydrolase (CD38) significantly increased by 94% (*p* < 0.05), 170% (*p* < 0.001), 470% (*p* < 0.001) and 640% (*p* < 0.001) in the HLU group compared to the CON group, respectively (Figure 4C–E,I). Meanwhile, the expression levels of Na^+^/Ca^2+^ exchanger 3 (NCX3) and Ca^2+^/calmodulin dependent protein kinase II (CaMK II) significantly reduced by 54% (*p* < 0.001) and 33% (*p* < 0.01) in the HLU group compared to the CON group, respectively (Figure 4F,G). The expression level of Protein kinase A (PKA) presented no significant change (*p* > 0.05) between the CON and HLU groups (Figure 4H). The sampling order and details of WB were presented in Appendix A.

### 3.4. The IP_3_ and IP_4_ Content in Muscle

Compared with the CON group, the IP_3_ content increased by 12% (*p* < 0.05) and IP_4_ content presented no significant change (*p* > 0.05) in the HLU group (Figure 5A,B).

### 3.5. Apoptosis of Myonucleus and Involvement of DNase X

It has been reported that PCM1 staining can label the myonucleus [27]. We had performed the PCM1 and dystrophin immunofluorescence double staining on skeletal muscle tissue sections, and then confirmed that the PCM1 staining can specifically label the myonucleus (the detailed results were presented in Appendix A). In this study, PCM1 and TUNEL immunofluorescence double staining was used to detect myonuclear apoptosis. The double staining and merge are shown in Figure 6A. Co-localization results in the CON and HLU groups were shown in Figure 6B. The co-localization results showed that the number of apoptotic total nuclei significantly increased from 0.3% ± 0.01% in the CON group to 4.6% ± 0.4% in the HLU group (*p* < 0.001). The number of apoptotic myonuclei also significantly increased from 20.2% ± 1.4% in the CON group to 51.8% ± 1.6% in the HLU group (*p* < 0.001) (Figure 6C).

PCM1 and DNase X immunofluorescence double staining was used to detect the location of DNase X. The double staining and merge in the CON and HLU groups were shown in Figure 7. The co-localization in Merge indicated that most of DNase X located in the myonuclei.

TUNEL and DNase X immunofluorescence double staining was used to detect the presence of DNase X in apoptotic nuclei. The double staining and merge in the CON and HLU groups were shown in Figure 8A. The co-localization results showed that the percentage of TUNEL-positive nuclei co-localized with DNase X to TUNEL-positive nuclei was 16.2% ± 2.6% in the CON group, which significantly increased to 61.1% ± 2.2% in the HLU group (*p* < 0.001) (Figure 8B). The negative control for Immunofluorescence were presented in Appendix A.

### 3.6. The DNase X Activity

Zn^2+^ had been reported to effectively inhibit DNase X activity [28], The Zn^2+^ was added into muscle extraction to testify whether the muscle extraction could successfully digest the plasmid DNA or not. As shown in Figure 9A, the DNA fragments below 1000 bp in length were rich in lane 2, but rare in lanes 3 and 4. It indicated that the plasmid DNA was indeed digested by the muscle extraction. According to the previous report that free Ca^2+^ was required for DNase X activation [16], EGTA was added into muscle extraction to testify the dependence of DNase X activity on free Ca^2+^. In Figure 9B, the DNA fragments below 1000 bp were rare in lane 1–6. It indicated that the activity of DNase X extracted from the CON or HLU group depended on free Ca^2+^. In Figure 9C, the DNA fragments below 1000 bp in lane 4–6 were richer than those in lane 1–3. The level of DNA fragmentation (the percentage of DNA fragments below 1000 bp to total DNA fragments) was expressed in Figure 9D. The level of DNA fragmentation in the HLU group was significantly higher than that in the CON group. It implied that the DNase X in the HLU group was more active than that in the CON group.

## 4. Discussion

In this study, as a model of hindlimb unloading, rats were suspended by their tails for 14 days. Severely disuse-atrophied SOL muscle was selected to investigate changes in nuclear Ca^2+^ levels, as well as the mechanism underlying these changes and the relationship with nuclear apoptosis. Our findings are as follows:

### 4.1. Hindlimb Unloading Leads to [Ca^2+^]_NE_ and [Ca^2+^]_NC_ Elevating in SOL Muscle

The nuclear envelope lumen, which is connected to the endoplasmic reticulum, is regarded as an intracellular Ca^2+^ pool. The nuclear envelope is surrounded by the INM and ONM, which store Ca^2+^ as a source for the nucleocytoplasm [29]. Therefore, [Ca^2+^]_NE_ levels may affect [Ca^2+^]_NC_ levels. As such, we used two Ca^2+^ fluorescent dyes to detect the [Ca^2+^]_NE_ and [Ca^2+^]_NC_ levels, respectively. Ca^2+^ fluorescent dye Cal-520 AM is a methyl acetyl derivative that can penetrate the membrane to combine with free Ca^2+^ in the nuclear envelope, while Ca^2+^ fluorescent dye Cal-520 Dextran can pass directly through the nuclear pores into the nucleocytoplasm, where it accumulates and remains for a long time to indicate [Ca^2+^]_NC_ levels [30]. Previous studies have used such Ca^2+^ fluorescent dyes (-AM and -Dextran) to detect [Ca^2+^]_NE_ and [Ca^2+^]_NC_ in the nuclei of liver cells [30], islet cells [31] and neurons [20]. Notably, the Cal-520 probe shows significant improvements in the signal-to-noise ratio and intracellular retention compared to other dyes. In the present study, we found that the [Ca^2+^]_NE_ and [Ca^2+^]_NC_ were significantly higher after disuse. Similar to previous studies and our findings that skeletal muscle disuse can lead to severe cytoplasmic Ca^2+^ overload [3,4,5], Ca^2+^ levels in the nucleus of skeletal muscle were also significantly elevated and nuclear Ca^2+^ overload may be associated in the development of disuse muscle atrophy.

In addition to the nucleus, mitochondria are also one of the primary organelles responsible for regulating intracellular Ca^2+^ homeostasis. Dysregulation of mitochondrial Ca^2+^ is a key mechanism leading to cytoplasmic Ca^2+^ overload. Notably, mitochondrial damage resulting from Ca^2+^ overload in mitochondrion has been observed in atrophic skeletal muscle [32]. Based on our findings, it is possible that Ca^2+^ overload is a common phenomenon that occurs in the main organelles of skeletal muscle cells after hindlimb unloading. Overall, disorder of structure and function of organelles due to Ca^2+^ overload may be an important reason for muscle atrophy.

### 4.2. Changes in Ca^2+^ Transport Capacity by INM and ONM may Explain why Ca^2+^ Levels Increase in Nuclear Envelope and Nucleocytoplasm during Hindlimb Unloading

#### 4.2.1. Ca^2+^ Uptake from the Cytoplasm on ONM Increases in Disused SOL Muscle

The ONM is the interface between the nuclear envelope and cytoplasm and contains several Ca^2+^ transporters proteins, including Ca^2+^-ATPase and inositol 1,3,4,5-tetrak-isphosphate receptor (IP_4_R). The Ca^2+^-ATPase located on the ONM is the same as that located on the endoplasmic reticulum membrane, and is a P-type ion pump that consumes adenosine triphosphate (ATP) to achieve active transport of Ca^2+^ from the cytoplasm to the nuclear envelope [33]. There are three isoforms of Ca^2+^-ATPase: type 1 is mainly expressed in fast muscle fiber, type 2 is mainly expressed in cardiomyocytes and slow muscle fiber, and type 3 is mainly expressed in non-muscle tissue cells [34]. As SOL muscle is primarily composed of slow muscle fiber, we measured the expression of Ca^2+^-ATPase type 2 (Ca^2+^-ATPase2) in the nuclei of SOL muscle. Results showed an increased expression level of nuclear Ca^2+^-ATPase2 in the disused SOL muscle. The finding suggests that the capacity of ONM to transport Ca^2+^ from the cytoplasm to nuclear envelope by Ca^2+^-ATPase2 may be enhanced under disuse conditions. 

IP_4_R on the ONM also can transport Ca^2+^, although under certain specific conditions. IP_4_R must first combine with IP_4_ and will open under high cytoplasmic Ca^2+^ concentrations (3–10 times higher than normal). When the IP_4_R channel opens, cytoplasmic Ca^2+^ enters into the nuclear envelope lumen along the Ca^2+^ concentration gradient [35]. Here, we measured IP_4_ content in the SOL muscle, but found no change after hindlimb unloading. These findings imply that IP_4_R activity does not change after skeletal muscle disuse. Previous studies have shown that skeletal muscle cytoplasmic Ca^2+^ levels would increase by about 1.1 times after disuse [4,5]. Such an increase may not be enough to initiate IP_4_R opening. Therefore, IP_4_R on the ONM may not be involved in the entry of cytoplasmic Ca^2+^ into the nuclear envelope.

Overall, the marked increase in Ca^2+^-ATPase2 expression on the ONM, resulting in the enhanced Ca^2+^ uptake from the cytoplasm, was involved in the increase in [Ca^2+^]_NE_ in skeletal muscles after hindlimb unloading.

#### 4.2.2. Ca^2+^ Release Increases while Ca^2+^ Uptake Attenuates on INM in Disused Muscle

The INM regulates [Ca^2+^]_NC_ by releasing Ca^2+^ from the nuclear envelope into the nucleocytoplasm and recycling it back. Two Ca^2+^ releasing channels are located on INM, the ryanodine receptor (RyR) and inositol 1,4,5-tetrakisphosphate receptor (IP_3_R), which releases free Ca^2+^ from nuclear envelope into nucleocytoplasm. There are at least three subtypes of RyR: RyR1 (in skeletal muscle), RyR2 (in cardiac muscle), and RyR3 (mainly in brain) [36]. RyR channel opens as binding with cyclic ADP-ribose (cADPR), which is produced in the nucleus by the transformation of NAD(P)+ by Cyclic ADP ribose hydrolase (CD38) [37]. Kraner et al. found that RyR1 expression increased in sarcoplasmic reticulum of hindlimb unloading rat SOL muscle [38]. In the present study, we further found that the expression of RyR1 and CD38 increased in SOL muscle nucleus after hindlimb unloading. It implies that the Ca^2+^ release from nuclear envelope into nucleocytoplasm is enhanced in skeletal muscle cell after hindlimb unloading. IP_3_R channel (subtype 1, IP_3_R1) is also located on the INM and release Ca^2+^ from nuclear envelope after binding with IP_3_ [24,39]. IP_3_R plays an important role in controlling nuclear Ca^2+^ level. For example, persistent atrial fibrillation (AF) can lead to nuclear Ca^2+^ overload in atrial muscle due to the elevation of nuclear IP_3_R expression, whereas IP_3_R knockdown could prevent AF-induced nuclear Ca^2+^ overload [40]. In our study, both IP_3_ content and IP_3_R1 expression markedly increased in the SOL muscle after hindlimb unloading. These changes suggest that the Ca^2+^ release from nuclear envelope via IP_3_R1 also increase. Based on the above findings, we think that the enhancement of releasing Ca^2+^ through Ca^2+^ channels on INM is an important cause of [Ca^2+^]_NC_ overload after muscle disuse.

Meanwhile, we have measured the expression of Ca^2+^/calmodulin dependent protein kinase II (CaMK II) and protein kinase A (PKA) in SOL muscle nucleus. CaMK II and PKA can phosphorylate the RyR1 and IP_3_R1 channels, respectively. This phosphorylation does not open the channels directly, but rather increases the opening probability of the activated channels [41,42]. To our surprise, the expression levels of CaMK II and PKA were reduced in SOL muscle after hindlimb unloading, which would inhibit channel opening probability and weaken Ca^2+^ release from nuclear envelope. Inhibit the opening probability of channel while increasing the content of channel protein in the skeletal muscle nucleus after hindlimb unloading. We speculate that such opposite changes may be a protective mechanism against excessive increases in nucleocytoplasmic Ca^2+^.

In addition to releasing Ca^2+^ into nucleocytoplasm, the INM recycles Ca^2+^ from nucleocytoplasm back to nuclear envelope, performed via Na^+^/Ca^2+^ exchanger (NCX) [8,43]. Previous research in mice has shown that reduced nuclear NCX activation in cerebellar neurons can lead to an increase in nucleocytoplasmic Ca^2+^ concentration, while reactivation of NCX can recovery nuclear Ca^2+^ concentration [44]. Nuclear NCX appears to play an important role in maintaining nuclear Ca^2+^ homeostasis. The mammalian NCX family consists of three subtypes: NCX1 is found in neurons, glial cells, cardiomyocytes, and kidney tissues; NCX2 is mainly in brain and NCX3 is mainly in skeletal muscle [45]. We found that NCX3 expression was reduced after hindlimb unloading, which would attenuate Ca^2+^ uptake from nucleocytoplasm to nuclear envelope. This may also be one of the reasons for increasing nucleocytoplasmic Ca^2+^ overload.

In summary, the enhanced Ca^2+^ uptake from cytoplasm into nuclear envelope on ONM, the enhanced Ca^2+^ release from nuclear envelope into nucleocytoplasm, and the attenuated Ca^2+^ recycle from nucleocytoplasm on INM are involved in the Ca^2+^ overload in the skeletal muscle nuclear envelope and nucleocytoplasm after hindlimb unloading.

### 4.3. Nucleocytoplasmic Ca^2+^ Overload Leads to Myonuclear Apoptosis by Increasing DNase X Activity during Hindlimb Unloading

Nuclear Ca^2+^ regulates gene transcription and other processes, and the imbalances in nuclear Ca^2+^ homeostasis, would lead to some negative, even fatal, effects [46]. The DNA fragmentation increasing attributed to nucleocytoplasmic Ca^2+^ overload has been proved [47]. DNA breakage is thought to be associated with DNase I family. To date, four different types of DNase I family have been identified, including DNase I, DNase X, DNase γ, and DNAS1L2 [48]. DNase X is predominantly found in skeletal muscle and cardiac myocytes, and its activity increases with elevated levels of Ca^2+^ [22]. In our present study, we identified notable nuclear apoptosis occurring in rat SOL muscle after hindlimb unloading, and the nuclear apoptosis including myonuclear and non-myonuclear apoptosis. We examined the localization of DNase X and confirmed its predominance in myonuclei. Furthermore, we found that the proportion of apoptotic nuclei containing DNase X significantly increased in the SOL muscle after hindlimb unloading. These findings suggest that DNase X may be involved in nuclear apoptosis, especially myonuclear apoptosis in skeletal muscle after hindlimb unloading. We also confirmed that DNase X activity was significantly elevated after hindlimb unloading. This elevated activity can increase the risk of DNA damage, leading to nuclear apoptosis. Previous studies have shown that DNase X can be activated by even sub-micromolar increases in Ca^2+^ [49]. Therefore, we conclude that both myonuclei and non-myonuclei in rat SOL muscle exhibit notable apoptosis after 14 days of hindlimb unloading. Moreover, the myonuclear apoptosis may be related to the increase in nucleocytoplasmic Ca^2+^, leading to an enhancement in DNase X activity, which in turn triggering DNA damage.

Currently, debate exists regarding whether myonuclei are lost during muscle atrophy. While some studies reported that myonuclei were eliminated [50,51], others suggested they were not lost [52,53]. Some studies have pointed out that the discrepancies may be attributed to multiple factors, such as variations in experimental models, limited number of studies, and differences in research design [54]. It is important to note that the maintenance of quantity of myonuclei requires a balance between their production and apoptosis. Our findings demonstrated that the apoptosis occurs in myonuclei during hindlimb unloading; however, it remains unclear whether the total number of myonuclei is impacted by this process. Further research is required to address this question.

## 5. Conclusions

In the present study, we focused on the changes in nuclear calcium level in skeletal muscle after hindlimb unloading, its mechanisms and possible effects. Our findings are presented in Figure 10. We have proved that the Ca^2+^ level in both nuclear envelope and nucleocytoplasm significantly increases in atrophic SOL muscle after unloading. The expression of nuclear Ca^2+^-ATPase2 increases to enhance the Ca^2+^ uptake capacity by outer nuclear membrane, contributing to the increase in nuclear envelope Ca^2+^ level. The increased expression of RyR1 and IP_3_R1, as well as the increased levels of CD38 and IP_3_, may enhance the ability of inner nuclear membrane to release Ca^2+^ into nucleocytoplasm. Meanwhile, the decreased expression of nuclear NCX3 reduces the ability of inner nuclear membrane to restore nucleocytoplasmic Ca^2+^. These changes thereby result in the rising of nucleocytoplasmic Ca^2+^ level. The nucleocytoplasmic Ca^2+^ overload activates DNase X to degrade DNA, leading to myonuclear apoptosis in hindlimb unloaded skeletal muscle. This study explores the intrinsic causes and results of nuclear Ca^2+^ imbalance in disused skeletal muscle and explains the mechanisms in the occurrence of nuclear apoptosis. It lays a theoretical foundation for the screening of drug targets based on nuclear calcium regulation and for the development of prevention and treatment measures for disuse muscle atrophy.

## 6. Limitations

Firstly, the nuclear Ca^2+^ leakage maybe happened during nucleus isolation attributed to the ionic permeability of nuclear membrane, which would affect the accuracy of the measurement on the content of nuclear Ca^2+^. Yet, the aim of the present study was to detect the relative difference of nuclear Ca^2+^ levels in experimental and control groups. The same isolation processes may result in roughly the same extent of the nuclear Ca^2+^ leakage in the two groups. Moreover, we have carried out a confirmatory experiment, which detecting the phosphorylation levels of CaMK II to confirm the changes of nuclear Ca^2+^ levels [5]. The results also supported the conclusions in this manuscript. (As supporting evidence, the experiment was presented in Appendix A). To sum up, we think that the possible leakage of nuclear Ca^2+^ may have an insubstantial effect, and our results can still indicate the changes of nuclear Ca^2+^ levels in this study. Nonetheless, the absence of avoiding nuclear Ca^2+^ leakage still is a limitation of this study. Secondly, we did not assess whether the nuclear Ca^2+^ levels were in the linear indication range of the Ca^2+^ indicator Cal-520 AM in this study. Although the results had shown the obvious differences of calcium fluorescence intensity within individuals and groups, this indicator may not be the optimal choice. The appropriate Ca^2+^ indicator and the necessary measures to prevent nuclear Ca^2+^ leakage should be found out in future study.

## Figures and Tables

**Figure 1 cells-12-01077-f001:**
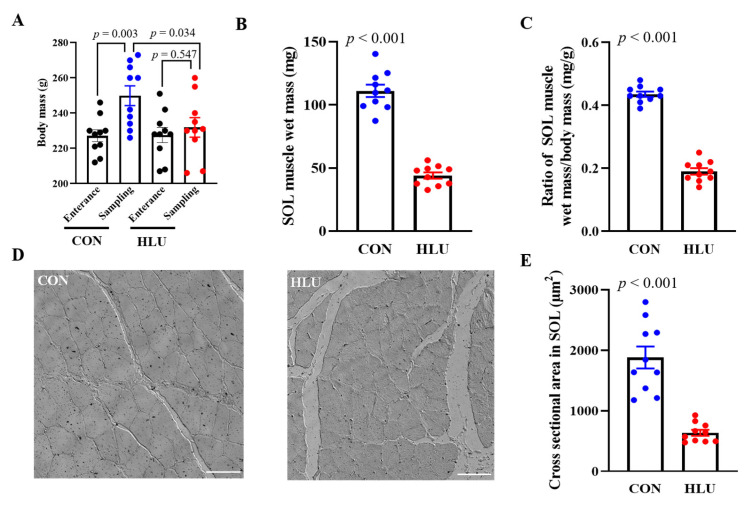
Changes of body and muscle mass and muscle cross-sectional area. (**A**) Changes of body mass. (**B**) Changes of SOL muscle wet mass. (**C**) Changes in ratio of SOL muscle wet mass/body mass. (**D**) Transection image of muscle fiber. (**E**) Changes in cross-sectional area (CSA). Each circle represented a value. *n* = 10. Data were analyzed by *t*-test. Data are shown as Mean ± SEM and considered statistically significant at *p* < 0.05. Scale bar, 60 μm.

**Figure 2 cells-12-01077-f002:**
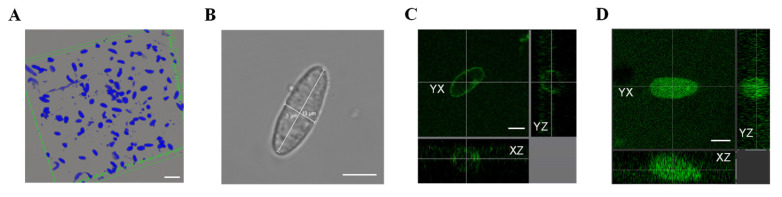
Morphology and free Ca^2+^ fluorescence staining of isolated nucleus. (**A**) Isolated nuclei were stained by DAPI. Scale bar, 30 μm. (**B**) Single isolated myonucleus under bright field. Length and width were measured as shown. Scale bar, 5 μm. (**C**) Free Ca^2+^ in nuclear envelope showed green fluorescence after loading Cal-520 AM (excitation, 493 nm; emission, 515 nm). (**D**) Free Ca^2+^ in nucleocytoplasm showed green fluorescence after loading Cal-520 Dextran (excitation, 493 nm; emission, 515 nm). Scale bar, 5 μm. To observe the distribution of Ca^2+^ fluorescence from more directions, the one myonucleus was observed in three orthogonal planes (YX plane, YZ plane, and XZ plane, as indicated by crossed dashed lines).

**Figure 3 cells-12-01077-f003:**
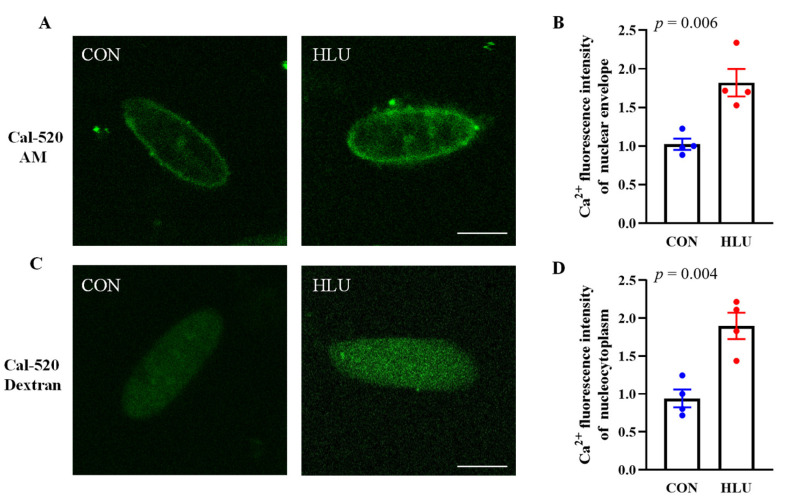
Free Ca^2+^ levels in nuclear envelope and nucleocytoplasm. (**A**) Free Ca^2+^ in nuclear envelope was showed by loading Cal-520 AM. (**B**) Change in Ca^2+^ fluorescence intensity in nuclear envelope. (**C**) Free Ca^2+^ in nucleocytoplasm was showed by loading Cal-520 Dextran. (**D**) Change in Ca^2+^ fluorescence intensity in nucleocytoplasm. Each circle represented a value. Observe at least 10 nuclei per individual. *n* = 4. Data were analyzed by *t*-test. Data are shown as Mean ± SEM and considered statistically significant at *p* < 0.05. Scale bar, 5μm.

**Figure 4 cells-12-01077-f004:**
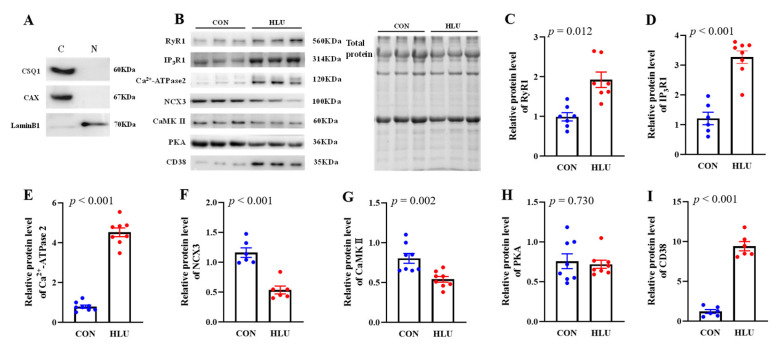
Relative expression level of Ca^2+^ transporters located on nuclear membrane and its regulatory proteins. (**A**) Representations of Calsequestrin 1 (CSQ1), Calnexin (CAX), Lamina protein B1 (LaminB1) in cellular protein fraction and nuclear protein fraction. C: cellular protein fraction; N: nuclear protein fraction. (**B**) Bands of Ca^2+^ transporters and its regulatory proteins. (**C**–**I**) Relative expression levels of Ryanodine receptor 1 (RyR1), Inositol 1,4,5-tetrakisphosphate receptor 1 (IP_3_R1), Ca^2+^-ATPase type 2 (Ca^2+^-ATPase2), Na^+^/Ca^2+^ exchanger 3 (NCX3), Ca^2+^/calmodulin dependent protein kinase II (CaMK II), Protein kinase A (PKA) and Cyclic ADP ribose hydrolase (CD38), respectively. Each circle represented a value. *n* = 6–8. Data were analyzed by *t*-test. Data are shown as Mean ± SEM and considered statistically significant at *p* < 0.05.

**Figure 5 cells-12-01077-f005:**
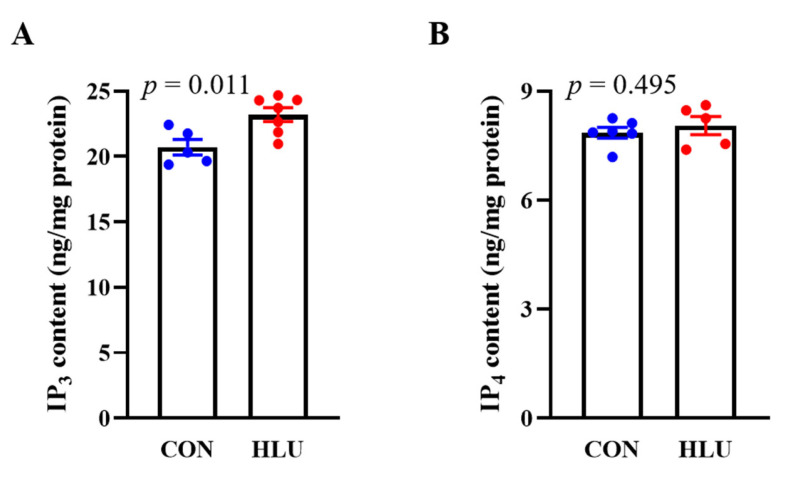
Inositol 1,4,5-tetrakisphosphate (IP_3_) and Inositol 1,3,4,5-tetrakisphosphate (IP_4_) content in muscle. (**A**) Content of IP_3_. (**B**) Content of IP_4_. Each circle represented a value. *n* = 5–7. Data were analyzed by *t*-test. Data are shown as Mean ± SEM and considered statistically significant at *p* < 0.05.

**Figure 6 cells-12-01077-f006:**
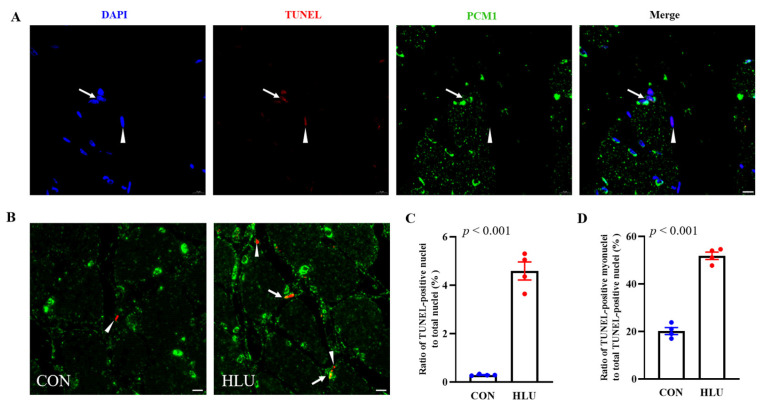
Ratio of TUNEL-positive nuclei and TUNEL-positive nuclei co-localized with pericentriolar material 1 (PCM1). (**A**) TUNEL assay and PCM1 immunofluorescence staining on muscle slice. Nuclei were blue after DAPI staining. TUNEL-positive nuclei were red after TUNEL staining. PCM1 was green after immunofluorescence staining. Nuclei labeled with PCM1 were myonuclei. TUNEL-positive nuclei co-localized with PCM1 (apoptotic myonuclei) were indicated by an arrow, TUNEL-positive nuclei not co-localized with PCM1 (apoptotic non-myonuclei) were indicated by an arrowhead in Merge. (**B**) Co-localization results of TUNEL assay and PCM1 immunofluorescence staining on SOL muscle slice in the CON and HLU groups. (**C**,**D**) Percentage of TUNEL-positive nuclei to total nuclei, and percentage of TUNEL-positive nuclei co-localized with PCM1 to total TUNEL-positive nuclei. Each circle representde a value. *n* = 4. Data were analyzed by *t*-test. Data are shown as Mean ± SEM and considered statistically significant at *p* < 0.05. Scale bar, 10 μm.

**Figure 7 cells-12-01077-f007:**
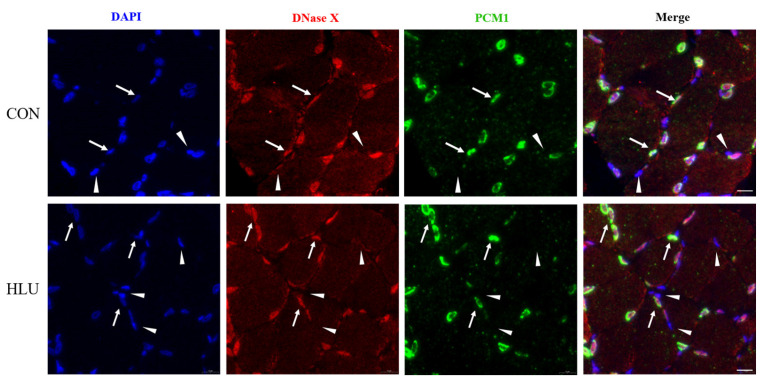
Co-localization of DNase X with PCM1. DNase X and pericentriolar material 1 (PCM1) immunofluorescence double staining on muscle slice in the CON and HLU groups. Nuclei were blue after DAPI staining. DNase X was red and PCM1 was green after immunofluorescence staining. DNase X co-localized with PCM1 was indicated by an arrow, and DNase X not co-localized with PCM1 was indicated by an arrowhead. Scale bar, 10 μm.

**Figure 8 cells-12-01077-f008:**
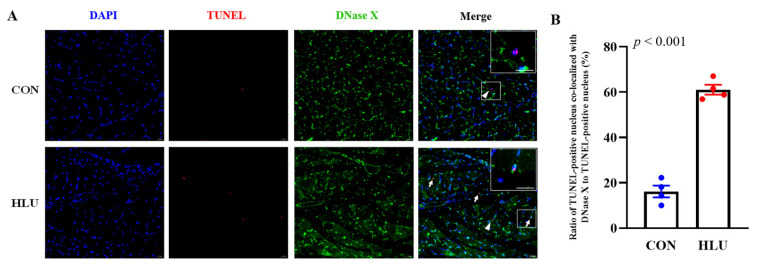
Ratio of TUNEL-positive nuclei co-localized with DNase X. (**A**) TUNEL assay and immunofluorescence staining on muscle slice in the CON and HLU groups. Nuclei were blue after DAPI staining. TUNEL-positive nuclei were red after TUNEL staining. DNase X was green after immunofluorescence staining. TUNEL-positive nuclei co-localized with DNase X were indicated by an arrow, TUNEL-positive nuclei not co-localized with DNase X were indicated by an arrowhead in Merge. Rectangular box view in upper right corner was magnified view of selected area in the picture. (**B**) Percentage of TUNEL-positive nuclei co-localized with DNase X to total TUNEL-positive nuclei. Each circle represented a value. *n* = 4. Data were analyzed by *t*-test. Data are shown as Mean ± SEM and considered statistically significant at *p* < 0.05. Scale bar, 20 μm.

**Figure 9 cells-12-01077-f009:**
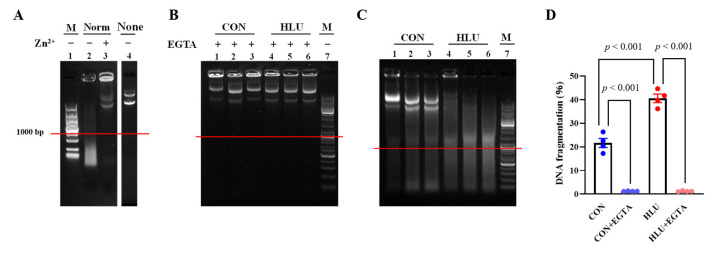
Activity of DNase X. (**A**) Lane 1: DNA Ladder (DNA molecular weight marker). Lane 2: Digestion solution which digested by muscle extraction (normal SOL muscle was extracted). Lane 3: Digestion solution which digested by the same muscle extraction as that in Lane 2 but containing 0.1 mM ZnCl_2_. Lane 4: Digestion solution which was digested by blank extraction (no muscle was extracted). (**B**) Lane 1–3: Digestion solution which was digested by CON-SOL muscle extraction containing 1 mM EGTA. Lane 4–6: Digestion solution which digested by HLU-SOL muscle extraction containing 1 mM EGTA. Lane 7: DNA Ladder. (**C**) Lane 1–6: Digestion solution which digested by the same muscle extraction as that in Picture B but without EGTA. Lane 7: DNA Ladder. The red line was the 1000 bp indicator line. (**D**) Levels of DNA fragmentation Each circle represented a value. *n* = 4. Data were analyzed by *t*-test. Data are shown as Mean ± SEM and considered statistically significant at *p* < 0.05.

**Figure 10 cells-12-01077-f010:**
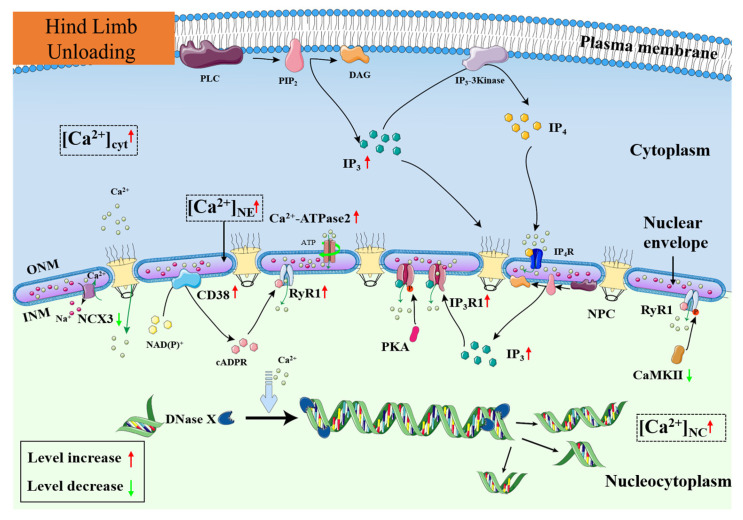
Schematic representation of the regulation changes of nuclear Ca^2+^ after hindlimb unloading. ATP: Adenosine triphosphate; Ca^2+^: Calcium ion; [Ca^2+^]_NC_: Ca^2+^ level in nucleocytoplasm; [Ca^2+^]_NE_: Ca^2+^ level in nuclear envelope; [Ca^2+^]_cyt_: Ca^2+^ level in cytoplasm; cADPR: Cyclic ADP-ribose; CaMK II: Ca^2+^/calmodulin dependent protein kinase II; CD38: Cyclic ADP ribose hydrolase; DAG: Diacylglycerol; DNase X: Deoxyribonuclease X; INM: Inner nuclear membrane; IP_3_: Inositol 1,4,5-tetrakisphosphate; IP_4_: Inositol- 1,3,4,5-tetrakisphosphate; IP_3_K: Inositol 1,4,5-tetrakisphosphate Kinase; Na^+^: Sodium ion; NCX3: Na^+^/Ca^2+^ exchanger 3; NE: Nuclear envelope; NPC: Nuclear pore complex; ONM: Outer nuclear membrane; PKA: Protein Kinase A; PIP_2_: Phosphatidylinositol (4,5) bisphosphate; PLC: Phospholipase C; RyR1: Ryanodine receptor 1.

**Table 1 cells-12-01077-t001:** Primary antibodies used in Western blot assay.

Protein Name	Antibody Details
Ca^2+^-ATPase type 2 (Ca^2+^-ATPase2)	1:1000, 4388S, Cell Signaling Technology, Danvers, MA, USA
Ca^2+^/calmodulin dependent protein kinase (CaMK Ⅱ)	1:1000, DF2907, Affinity Biosciences, Cincinnati, OH, USA
Calnexin (CAX)	1:1000, YT0613, Immunoway, Plano, TX, USA
Cyclic ADP ribose hydrolase (CD38)	1:1000, YT5392, Immunoway, Plano, TX, USA
Calsequestrin 1 (CSQ1)	1:1000, ab191564, Abcam, Cambridge, UK
Inositol 1,4,5-tetrakisphosphate receptor (IP_3_R1)	1:1000, ab108517, Abcam, Cambridge, UK
Lamina protein B1 (LaminB1)	1:1000, 12586S, Cell Signaling Technology, Danvers, MA, USA
Na^+^/Ca^2+^ exchanger 3 (NCX3)	1:1000, YN1335, Immunoway, Plano, TX, USA
Protein kinase A (PKA)	1:1000, AF5450, Affinity Biosciences, Cincinnati, OH, USA
Ryanodine receptor 1 (RyR1)	1:1000, 8153S, Cell Signaling Technology, Danvers, MA, USA

## Data Availability

The data presented in this study are available on request from the corresponding authors.

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
