# Peer review of "New Findings: Hindlimb Unloading Causes Nucleocytoplasmic Ca2+ Overload and DNA Damage in Skeletal Muscle"

_cells, 2023, doi:10.3390/cells12071077_

Round 1

Reviewer 1 Report

In this study, rat hindlimb suspension for 14 days was used as a model of disuse muscle atrophy. It is noteworthy that the authors succeeded in quantifying changes in nuclear Ca2+ concentration in the severely atrophied soleus muscle (SOL).

In particular, the quantification of Ca2+ concentration in each region of the nucleus is a highly original finding. Furthermore, the authors quantified relevant proteins that explain the lack of Ca2+ homeostasis in the nucleus. These findings on disuse atrophy are of scientific originality.

On the other hand, there are some major concerns regarding the method used to quantify myocyte nuclei.

Skeletal muscle tissue contains satellite cells, vascular endothelial cells, smooth muscle cells, and nerve cell nuclei in addition to myocyte nuclei. Therefore, it should be proven that the extracted nuclei are myocyte nuclei. Identification of myocyte nuclei by histochemistry should also confirm that the nuclei were present inside the plasma membrane. The same perspective should be taken with regard to biochemical data.

Skeletal muscle apoptosis is also known to be regulated by mitochondria. Moreover, mitochondria are the major regulatory organelle of intracellular Ca2+ homeostasis. Ca2+ in skeletal muscle is regulated by organelle coordination, including sarcoplasmic reticulum and calcium. Therefore, the involvement of mitochondria should at least be mentioned in the discussion.

Minor points

Pentobarbital is used for anesthesia. What is the reason for using this anesthetic?

Fig.2  Scales should be provided for A, C-D photographs.

Fig.3  Morphological data of the nuclei should be added.

Fig.6  It is necessary to evaluate whether the nucleus is a myocyte nucleus.

Reviewer 2 Report

Yang et al analyzed the Ca2+ distribution in the myonuclear envelope and nucleoplasm in nuclei isolated from rat skeletal muscles subjected to disuse induce atrophy. They observed an increase in Ca2+ concentration both in the nuclear envelope and in the nucleoplasm of the atrophic nuclei. This correlates with an increased expression of endoplasmic/reticulum Ca2+-related protein in the nuclear envelope and an increased amount of inositol-3-phosphate. The increased nuclear Ca2+ concentration is accompanied by the increased number of TUNEL-positive nuclei and the increased activity of DNAse X. The authors conclude that the increase in the nuclear Ca2+ concentration can contribute to nuclear apoptosis in disuse muscle atrophy. The work has a linear flow but it addresses naively the Ca2+ measurements and the muscle pathophysiology. Different major concerns arise:

1.       The authors measured Ca2+ levels in isolated myonuclei. Since the nuclear envelope is highly permeable to ions, it is not clear how the Ca2+ concentration in the nucleoplasm is preserved during the isolation protocol. Furthermore, in the materials and methods section, the buffer composition used during the measurements is not reported. In my opinion, the only way to perform reliable measurement of Ca2+ level in the nucleoplasm is in intact fibres loaded with Ca2+ dye.

2.       To measure Ca2+ level in the nuclear envelope, the authors used the Ca2+ indicator Cal-520 AM. This indicator has a high affinity for Ca2+ (Kd=320 nM) and it is normally used to measure Ca2+ in the cytosolic compartment where Ca2+ concentration is almost 100 times less than in the store-compartments. Since the nuclear envelope is a continuation of the endoplasmic reticulum, the expected Ca2+ concentration should be around 100 µM, the concentration at which high-affinity dyes normally reach saturation. Did the author assess if the measurement is still in the linear range of the Ca2+ indicator? Furthermore, the nuclei are maintained in a solution without Ca2+, is it possible that the nuclear envelope leak Ca2+ in these experimental conditions? Also, in this case, the only way to perform reliable measurement of Ca2+ level in the nucleoplasm is in intact fibres loaded with Ca2+ dye.

3.       The authors analysed the activity and localization of DNAse X in the TUNEL-positive myonuclei. The role of DNAse X and its localization is not completely elucidated, but some reports localized this DNAse in the extracellular compartment and the secretory pathway (DOI: 10.1074/jbc.M610428200). Also, the role of this DNAse in apoptosis is not clearly reported. In my opinion, the link between this observation and the increase in apoptosis nuclei is weak. First, did the authors verify the localization of DNAse X in the muscle sections? I suggest performing a co-staining with an antibody against a membrane protein to check the localization of DNAse X. Furthermore, why the authors excluded all the other DNAse family members is not clear.

4.       There is no consensus about the apoptosis role in muscle atrophy. Indeed, recent reports demonstrated that during atrophic stimuli the number of nuclei in muscle fibres remains constant (DOI: 10.1016/j.yexcr.2011.05.013, DOI: 10.1073/pnas.0913935107). Furthermore, the myonuclear domain hypothesis become to be controversial, since it has been demonstrated that apoptotic stimulation propagates across the cell in multinucleated cells (DOI: 10.1016/j.placenta.2012.09.013). In my opinion, the authors conclusions are not supported by the data. Did the authors check if the TUNEL-positive nuclei are nuclei of myofibers? Also, in this case, I suggest performing a co-staining with an antibody against a membrane protein to check the localization of TUNEL-positive nuclei.

Round 2

Reviewer 1 Report

The authors have sufficiently address my previous comments. Thank you. 

Author Response

Dear Reviewer,

     We greatly thank the Reviewers for their kindly comments and valuable suggestions.